# Mathematical Modeling and Robustness Analysis to Unravel COVID-19 Transmission Dynamics: The Italy Case [note 1]

**DOI:** 10.3390/biology9110394

**Published:** 2020-11-11

**Authors:** Chiara Antonini, Sara Calandrini, Fabrizio Stracci, Claudio Dario, Fortunato Bianconi

**Affiliations:** 1ICT4life srl, Via Mario Donati Guerrieri, 16, 06132 Perugia, Italy; sara.calandrini@i4l.company; 2Department of Engineering, University of Perugia, Via Goffredo Duranti, 93, 06125 Perugia, Italy; 3Department of Experimental Medicine, University of Perugia, Piazzale Settimio Gambuli, 06132 Perugia, Italy; fabrizio.stracci@unipg.it; 4Regional Government of Umbria, Corso Vannucci, 96, 06121 Perugia, Italy; cdario@regione.umbria.it; 5COVID-19 Epidemiological Unit, Regional Government of Umbria, Corso Vannucci, 96, 06121 Perugia, Italy; fortunato.bianconi@gmail.com

**Keywords:** COVID-19, SARS-CoV-2, SEIR model, Bayesian parameter estimation, conditional robustness analysis, Italy

## Abstract

**Simple Summary:**

In the absence of vaccines and antiviral therapies against COVID-19, non-pharmaceutical interventions represent the only weapon to fight this new coronavirus. In this context, mathematical models are an important tool for supporting the study of spread and transmission of this disease and to understand how restrictive measures could control the epidemic. In the current study, we adopt a new mathematical model representing the dynamics of COVID-19. We estimate model parameters with a new Bayesian method while using the public Italian data, in order to reproduce the pandemic evolution for Italy and one of its regions, Umbria. Once the model is calibrated, we also apply an algorithm, called Conditional Robustness Analysis. This algorithm allows for us to understand the influence of epidemiological parameters and non-pharmaceutical interventions on the number of hospitalized patients. This pipeline of analysis provides a quantitative explanation of the number of underestimated positive cases during the first wave of the epidemic and of the impact of lock-down measures on the hospitalization capacity. Moreover, the calibrated model is quite accurate for making updated estimations of the epidemic evolution.

**Abstract:**

This study started from the request of providing predictions on hospitalization and Intensive Care Unit (ICU) rates that are caused by COVID-19 for the Umbria region in Italy. To this purpose, we propose the application of a computational framework to a SEIR-type (Susceptible, Exposed, Infected, Removed) epidemiological model describing the different stages of COVID-19 infection. The model discriminates between asymptomatic and symptomatic cases and it takes into account possible intervention measures in order to reduce the probability of transmission. As case studies, we analyze not only the epidemic situation in Umbria but also in Italy, in order to capture the evolution of the pandemic at a national level. First of all, we estimate model parameters through a Bayesian calibration method, called Conditional Robust Calibration (CRC), while using the official COVID-19 data of the Italian Civil Protection. Subsequently, Conditional Robustness Analysis (CRA) on the calibrated model is carried out in order to quantify the influence of epidemiological and intervention parameters on the hospitalization rates. The proposed pipeline properly describes the COVID-19 spread during the lock-down phase. It also reveals the underestimation of new positive cases and the need of promptly isolating asymptomatic and presymptomatic cases. The results emphasize the importance of the lock-down timeliness and provide accurate predictions on the current evolution of the pandemic.

## 1. Introduction

The global spread of the COVID-19 respiratory syndrome is related to a new strain of coronavirus called severe acute respiratory syndrome coronavirus 2 (SARS-CoV-2). It has caused a large number of deaths worldwide and it has had significant economic costs for the most affected countries. The new SARS-CoV-2 pathogen was initially identified in the city of Wuhan, in the Hubei Province of China, in late December 2019 and, since then, it has rapidly spread to every corner of the globe. On 11 March 2020, the World Health Organization (WHO) declared the COVID-19 outbreak a pandemic [1].

Italy was one of the first European countries being severely hit by COVID-19. On 21 February 2020, first clusters of pneumonia cases were detected in Lombardy and Veneto while, by the beginning of March, the virus had already spread to all regions of Italy [2]. Given this scenario, the Italian government imposed multiple intervention measures, starting from the institutions of two ‘red areas’ in Lombardy and Veneto to a strict national lock-down from 11 March 2020 until 3 May 2020. In May, when the number of cases started to progressively decrease, most restrictions were gradually lifted and, beginning from June, movement between Italian regions and European countries was allowed. At the end of the summer, new positive cases started to increase again, which were mostly imported by people returning from vacations overseas. However, even though the situation is still alarming, it is currently under control, mostly thanks to a massive swab campaign for promptly detecting and isolating infected people.

When compared to previous coronavirus outbreaks, COVID-19 has shown some peculiar features, with presymptomatic and asymptomatic transmission being the most alarming one. Indeed, as shown in [3], infected people with no or mild symptoms release large amounts of virus in the early stage of infection. This aspect makes symptom-based isolation ineffective and poses new and difficult challenges for the containment of human-to-human transmission.

In this context, mathematical models represent a quantitative support for understanding the spread mechanism and the dynamic of the pandemic, providing estimates of the effectiveness of different interventions [4,5]. Because the diffusion of COVID-19 is highly complex, due to the many parameters and factors involved, different models have been proposed. In the context of epidemiology, two widely used compartmental models are the so called Susceptible-Infectious-Removed (SIR) and Susceptible-Exposed-Infectious-Removed (SEIR) models. For the Italian situation, one of the first epidemic models developed to describe the disease evolution is the one in [6]. In this work, a variant of the SIR model is presented and detected positives are related with the unknown number of actual infections through the definition of a new factor. In [5], the SIDARTHE model is explained, which distinguishes infected individuals based on diagnosis and illness severity. The work in [7] studies the COVID-19 epidemic as a feedback control problem, mainly focusing on the efficacy of suppression and mitigation strategies. On the other hand, in [8], Italy is modeled as a network of regions and the importance of differentiated, but coordinated feedback regional interventions is strongly highlighted.

This article started from the need to provide predictions and estimations on hospitalization and Intensive Care Unit (ICU) rates for the Health Government of the Umbria region in Italy, in order to support it in making decisions regarding staff resources and hospital beds [9]. Given this request, in order to represent the COVID-19 epidemic dynamics in Italy, we use a compartmental epidemiological model taken from [10], based on the classic SEIR model with lock-down (L) measures (SEIRL). A simplified version of the model is presented in [11], where it is used to analyze the consequences of household transmission in the evolution of the pandemic. In order to make the model representative of the Umbria scenario during the initial phase of the pandemic, we estimate model parameters against the public COVID-19 data of Umbria using our Bayesian method for model calibration, called Conditional Robust Calibration (CRC) [12,13,14]. Subsequently, we also perform a Conditional Robustness Analysis (CRA), through our CRA Toolbox, in order to understand the influence of epidemiological parameters and progressive restriction measures on the hospitalization capacity and logistics [15,16]. Through the analysis of the CRA results, we can make some considerations regarding the evolution of the pandemic in Umbria. Furthermore, we apply the same computational pipeline to the Italian scenario, as the national case study. Our analysis reveals how the combination of model calibration and robustness analysis is a useful framework for making reliable predictions on the scale of the outbreak and on the evaluation of the timescale and impact of non-pharmaceutical interventions.

## 2. Methods

### 2.1. Mathematical Model

Figure 1 shows the SEIRL model used in this work. It takes into account different clinical stages of the infection, requiring distinct levels of healthcare. Firstly, susceptible individuals (class *S*) are members population who risk to become infected. Those who are infected start out in an exposed (class *E*), where they develop the infection, but do not transmit it, since they do not eliminate the virus. However, after a certain period, *E* individuals enter the presymptomatic class (class $P_{S}$), since they may be able to transmit the virus before developing or not the symptoms. The rate of exit from *E* is $a_{0}$ and from $P_{S}$ is $a_{1}$. After leaving the class $P_{S}$, a fraction of them develops asymptomatic infection (class *A*) and then recovers (class *R*) at rate $g_{0}$. On the other hand, the remaining fraction of people begins with a mild infection (class *M*). People with mild infection have flu symptoms, like fever and cough, which do not require hospitalization. Subsequently, people in *M* class either recover from the infection (*R*), at rate $g_{1}$, or progress to a severe infection (class *H*), at rate $p_{1}$. These patients have severe pneumonia that typically requires hospitalization. Again, severe infection either recovers (*R*) at rate $g_{2}$ or progresses to critical infection (class ICU) at rate $p_{2}$. Individuals with critical infection have life-threatening symptoms, like respiratory failure, and they need treatment in an ICU. From ICU, people either recover (*R*) at rate $g_{3}$ or die (class *D*) at rate *u*. Recovered individuals are assumed to be protected from re-infection for life. The transmission rates at stage $P_{S}$, *A*, *M*, *H*, and ICU are described by parameters $b_{e}$, $b_{0}$, $b_{1}$, $b_{2}$, and $b_{3}$, respectively. All of the rates are per day. The model assumes that only people in a critical stage die and that all individuals have equal susceptibility to infection. We suppose that the virus transmission rates are orders of magnitudes faster than human birth or death rates, which are completely neglected in the model. For this reason, the total population size $N=S+E+P_{S}+A+M+H+ICU+R+D$ is supposed to be constant [7,17,18]. We initialize the model assuming one exposed person and we normalize over the population N and multiply by $10^{5}$, i.e., $S_{0}=10^{5}-E_{0}$, $E_{0}=\left(1/N\right)\cdot 10^{5}$, $P_{S,0}=0$, $A_{0}=0$, $M_{0}=0$, $H_{0}=0$, $ICU_{0}=0$, $R_{0}=0$, $D_{0}=0$.

The dynamical system is represented by the following system of Ordinary Differential Equations (ODEs):(1)$$\begin{array}{cc} \; & \dot{S}=-\left(b_{e}P_{S}+b_{0}A+b_{1}M+b_{2}H+b_{3}ICU\right)S\\ \; & \dot{E}=\left(b_{e}P_{S}+b_{0}A+b_{1}M+b_{2}H+b_{3}ICU\right)S-a_{0}E\\ \; & \dot{P_{S}}=a_{0}E-a_{1}P_{S}\\ \; & \dot{A}=fa_{1}P_{S}-g_{0}A\\ \; & \dot{M}=\left(1-f\right)a_{1}P_{S}-g_{1}M-p_{1}M\\ \; & \dot{H}=p_{1}M-g_{2}H-p_{2}H\\ \; & \dot{ICU}=p_{2}H-g_{3}ICU-uICU\\ \; & \dot{R}=g_{0}A+g_{1}M+g_{2}H+g_{3}ICU\\ \; & \dot{D}=uICU \end{array}$$

Variables of the ODEs are:*S*: susceptible individuals,*E*: exposed individuals,$P_{S}$: presymptomatic individuals,*A*: asymptomatic individuals,*M*: people with mild infection,*H*: people with severe infection which requires hospitalization,ICU: people with critical infection which requires ICU level care,*R*: individuals who have recovered from the disease, and*D*: dead people.

Parameters $b_{i},\;\forall i=e,0,1,2,3$ represent the transmission rates at the different stages of the disease, while parameters $g_{i}\;\forall i=0,1,2,3$ represent the different recovery rates and *u* the death rate. Parameters $a_{i}\;\forall i=0,1$ indicate the rate of exit from classes *E* and $P_{S}$, respectively. Parameters $p_{1}$ and $p_{2}$ are, respectively, the progression rate from mild to severe infection and from severe to critical infection. Because these rate constants are not generally measured directly in studies, they are related to the clinical observations through the following formulas:(1)$a_{1}=PresymPeriod^{-1}$(2)$a_{0}={\left(IncubPeriod-PresymPeriod\right)}^{-1}$(3)$g_{1}=DurMildInf^{-1}\cdot \left(1-FracSevere-FracCritical\right)$(4)$p_{1}=DurMildInf^{-1}-g_{1}$(5)$p_{2}=DurHosp^{-1}\cdot \frac{FracCritical}{\left(FracSevere+FracCritical\right)}$(6)$g_{2}=DurHosp^{-1}-p_{2}$(7)$u=TimeICUDeath^{-1}\cdot \frac{CFR}{FracCritical}$(8)$g_{3}=TimeICUDeath^{-1}-u$(9)f=FracAsym(10)$g_{0}=DurAsym^{-1}$,
where IncubPeriod is the incubation period, defined as the time between exposure to an infected person and the development of symptoms, while PresymPeriod is the length of the infectious phase of the incubation period. DurAsym is the average duration of asymptomatic infection, DurMildInf can be estimated as the duration of mild symptoms or the time from symptom onset to hospitalization for those who progress to a severe stage. The duration of severe infection DurHosp is the time from hospital admission to recovery or ICU admission. TimeICUDeath is the time from ICU admission to recovery or death. The CFR is the case fatality ratio that describes the fraction of all symptomatic infected individuals who die. All time durations are expressed in days (d). FracAsym is the percentage of infected people that develops asymptomatic infection, while FracSevere and FracCritical are, respectively, the percentage of individuals requiring hospitalization and ICU-level care. While using this model, the basic reproductive number $R_{0}$ is defined as the sum of:the average number of secondary infections generated from an individual in class $P_{S}$,the fraction of people that progresses to class *A* multiplied by the average number of secondary infections that are generated from an infected person in stage *A*,the fraction of people that progresses to class *M* multiplied by the average number of secondary infections generated from an infected person in stage *M*,the probability that an infected individual progresses to class *H* multiplied by the average number of secondary infections generated from a patient in stage *H*, andthe probability that an infected individual progresses to class ICU multiplied by the average number of secondary infections generated from a patient in stage ICU.

The formula of $R_{0}$ can be derived using the next generation matrix [10,19]:(2)$$R_{0}=N\left[\frac{b_{e}}{a_{1}}+f\frac{b_{0}}{g_{0}}+\left(1-f\right)\frac{1}{p_{1}+g_{1}}\left(b_{1}+\frac{p_{1}}{p_{1}+g_{2}}\left(b_{2}+b_{3}\frac{p_{2}}{u+g_{3}}\right)\right)\right].$$

Moreover, the model takes the possibility of an intervention into account in order to reduce infection probability. Thus, we introduce a parameter $s_{0}$ which represents the containment measures for decreasing presymptomatic and asymptomatic transmission. The ban of detected positive on leaving their houses is represented through parameter $s_{1}$. The employment of personal protective equipment (PPE) in hospitals also reduces the probability of infection, represented with parameter $s_{2}$. These parameters decrease the transmission rate constants, as follows:(3)$$\begin{array}{c} b_{e,lock}=b_{e}\cdot s_{0}\\ b_{0,lock}=b_{0}\cdot s_{0}\\ b_{1,lock}=b_{1}\cdot s_{1}\\ b_{2,ppe}=b_{2}\cdot s_{2}\\ b_{3,ppe}=b_{3}\cdot s_{2} \end{array}$$

The model is a positive and bilinear system: all state variables have non negative values for time t≥0, if initialized with non negative values at time 0. For the mass conservation property, we have that $\dot{S}+\dot{E}+\dot{P_{S}}+\dot{A}+\dot{M}+\dot{H}+\dot{ICU}+\dot{R}+\dot{D}=0$; hence, the sum of the states is constant and equal to the total population *N*. The system admits the following equilibrium: $S^{*}\geq 0$, $E^{*}=0$, $P_{S}^{*}=0$, $A^{*}=0$, $M^{*}=0$, $H^{*}=0$, $ICU^{*}=0$, $R^{*}\geq =0$, $D^{*}\geq =0$, with $S^{*}+R^{*}+D^{*}=1$. At the equilibrium, only susceptible, recovered, and dead individuals are present, meaning that the disease dies out. Thus, all points of equilibrium are given by $\left(S^{*},0,0,0,0,0,0,R^{*},D^{*}\right)$ with $S^{*}+R^{*}+D^{*}=1$. The system can be partitioned into three subsystems: the first one with only susceptible individuals, the second one with all of the infected and the third one which includes healed and dead. When infected individuals are zero, the other variables are at the equilibrium. Variable *S* is monotonically decreasing to $S^{*}$, while *R* and *D* are monotonically increasing to $R^{*}$ and $D^{*}$ if and only if all of the infected converge to zero. The system can be rewritten in feedback form with the infected subsystem as a positive linear subsystem subjected to a feedback signal *c*. Defining $x={\left[E\;P_{S}\;A\;M\;H\;ICU\right]}^{T}$, the subsystem is:(4)$$\dot{x}\left(t\right)=Fx\left(t\right)+dc\left(t\right)=\left[\begin{array}{cccccc} -a_{0} & 0 & 0 & 0 & 0 & 0\\ a_{0} & -a_{1} & 0 & 0 & 0 & 0\\ 0 & fa_{1} & -g_{0} & 0 & 0 & 0\\ 0 & \left(1-f\right)a_{1} & 0 & -(g_{1}+p_{1}) & 0 & 0\\ 0 & 0 & 0 & p_{1} & -(g_{2}+p_{2}) & 0\\ 0 & 0 & 0 & 0 & p_{2} & -(g_{3}+u) \end{array}\right]x\left(t\right)+\left[\begin{array}{c} 1\\ 0\\ 0\\ 0\\ 0\\ 0 \end{array}\right]c\left(t\right)$$
(5)$$c\left(t\right)=S\left(t\right)y_{S}\left(t\right)$$
(6)$$y_{S}\left(t\right)=\left[\begin{array}{cccccc} 0 & b_{e} & b_{0} & b_{1} & b_{2} & b_{3} \end{array}\right]x\left(t\right)$$
(7)$$y_{R}\left(t\right)=\left[\begin{array}{cccccc} 0 & 0 & g_{0} & g_{1} & g_{2} & g_{3} \end{array}\right]x\left(t\right)$$
(8)$$y_{D}\left(t\right)=\left[\begin{array}{cccccc} 0 & 0 & 0 & 0 & 0 & u \end{array}\right]x\left(t\right)$$
(9)$$\dot{S}\left(t\right)=-S\left(t\right)y_{S}\left(t\right)$$
(10)$$\dot{R}\left(t\right)=y_{R}\left(t\right)$$
(11)$$\dot{D}\left(t\right)=y_{D}\left(t\right)$$

It is possible to demonstrate that the feedback subsystem with susceptible population $S^{*}$ is asymptotically stable if and only if $S^{*}<\frac{1}{R_{0}}$. Moreover, for positive initial conditions, the limit value $S^{*}=\lim_{t\to +\infty }S\left(t\right)$ cannot exceed $\frac{1}{R_{0}}$ [5,20].

### 2.2. Data

The data for model calibration are available on the GitHub repository of the Italian Civil Protection Department [21]. We perform parameter estimation against hospitalized, ICU, and dead patients (H, ICU, and D in the model), since they represent the most reliable measures. Indeed, we suppose that the number of mild infections was underestimated during the initial phase of COVID-19 spread, due to under registration and testing only patients with symptoms or pneumonia [4,22].

### 2.3. Model Calibration Using CRC

CRC is a Bayesian methodology that belongs to the class of Approximate Bayesian Computation Sequential Monte Carlo (ABC-SMC) approaches. It considers the model parameter vector as a random variable **P**. Through perturbation of the parameter space, CRC aims at iteratively evaluating an approximation of the parameter posterior distribution conditioned to the available data $f_{P|y^{*}}\left(p\right)$, where $y^{*}$ is the dataset. The main steps of CRC are the following. First of all, we sample the parameter space P in order to generate $N_{S}$ parameter vectors inside the chosen lower and upper boundaries $L^{1}$ and $U^{1}$. The sampling technique adopted is Latin Hypercube Sampling (LHS), since it guarantees an even distribution of the points across the parameter space. The parameters are supposed to be uniformly or logarithmically distributed among their intervals. Defined as $P_{O}$ the matrix of generated parameter samples, for each $p\in P_{O}$, we integrate the ODE model in order to generate the vector of observables **y**. The output variables are then compared with the dataset through the definition of a distance function. In this work, we use the Absolute Distance Function (ADF) defined as the $l_{1}-norm$ sum of the distance between simulated and real data:(12)$$ADF_{i}=\sum_{j=1}^{k}\left|y_{i}\left(t_{j}\right)-y_{ij}^{*}\right|\;i=1,...,m,$$
where $y_{i}$ is the simulated observable at time point $t_{j}$ and $y_{ij}^{*}$ is the corresponding measured variable. Thus, for each **p**$\in P_{O}$, we compute $ADF_{i}\;i=1,...,m$ and only select those distance functions under a user defined threshold $\epsilon_{i}\geq 0$. At the end of this step, a subset $P_{O,\epsilon_{i}}\in P_{O}$ is defined for each output variable i=1,...,m. By intersecting all of these sets together, we obtain $P_{O,\epsilon }=\left\{{⋂}_{i=1}^{m}P_{O,\epsilon_{i}}\right\}$, where $\epsilon =\{\epsilon_{1},...,\epsilon_{i},...,\epsilon_{m}\}$. These accepted parameter samples guarantee all of the distance functions under all the chosen thresholds. Thus, they are used for estimation of the approximate posterior distribution $f_{P|P_{O,\epsilon }}$ through a kernel density approach. This calibration procedure can be repeated for multiple iterations until a user defined stopping criterion is met. In each subsequent iteration, the sampling interval is updated according to the posterior distribution of the previous iteration; typically, it is progressively shrunk around its mode vector $p_{mode}$. The output of the algorithm is $f_{P|P_{O,\epsilon }}$. More details on CRC can be found in [12,13,14]. The codes for running CRC and the ODE model of COVID-19 are available at https://github.com/fortunatobianconi/CRC.

### 2.4. Conditional Robustness Analysis

After model calibration, we perform the CRA on model parameters whlie using the conditional robustness algorithm implemented in the CRA Toolbox [15,16]. This method was initially developed in the field of Cancer Systems Biology for identifying those proteins responsible of the cancer proliferation activity. However, because it allows us to quantify how much the temporal behavior of a model variable is influenced by the perturbation of its parameters, it can easily be applied to other contexts. In this work, we are interested in evaluating the impact of model parameters on the hospitalization capacity, i.e. on the curve of H and ICU variables. First of all, the CRA defines an evaluation function $z_{i}$ for the chosen output node $y_{i}$, representative of the property of interest. The possible evaluation functions are the area under the curve of $y_{i}$ or its maximum value. Using LHS, the parameter space is perturbed and, for each perturbation, the model is integrated in order to compute the corresponding evaluation function. Then, the probability density function (pdf) of the evaluation function $f_{Z_{i}}\left(z_{i}\right)$ can be partitioned in two regions, which represent the lower and upper tail of the pdf:(13)$$L\left(\alpha \right)=\{z_{i}\leq a:\int_{0}^{a}f_{Z_{i}}\left(z_{i}\right)dz_{i}=\alpha \}$$
(14)$$U\left(\alpha \right)=\{z_{i}\geq a:\int_{a}^{\infty }f_{Z_{i}}\left(z_{i}\right)dz_{i}=\alpha \},$$
where α is the level of probability that represents the area under the lower and upper tail of the evaluation function pdf and *a* is the corresponding threshold value in the domain of the pdf. From this partition, we estimate the conditional pdfs of each parameter, i.e., $f_{p_{w}|L}$ and $f_{p_{w}|U}$, representing the distribution of the parameter vector when the evaluation function is in the lower or upper tail of $f_{Z_{i}}\left(z_{i}\right)$. The final result that is returned by the CRA is the Moment Independent Robustness Indicator (MIRI), defined as: (15)$$\mu_{t}=\int \left|f_{p_{w}|L}-f_{p_{w}|U}\right|dp_{w}\;w=1,...,q,$$
where *q* is the total number of model parameters. Parameters with a high MIRI value have a strong impact on the selected output variable since their conditional pdfs are highly separated, determining a significant variation of the evaluation function behavior (see Appendix A). More details on the CRA and on the CRA Toolbox can be found in [14,15,16] and the code is available at http://gitlab.ict4life.com/SysBiOThe/CRA-Matlab.

## 3. Results

### 3.1. CRC Results: Italy Case

We calibrate the SEIRL model while using the Italian data from 24 February to 3 May 2020. We suppose that the initial day of virus introduction is 30 days prior to the first ten registered deaths [22]. Moreover, we introduce the following interventions:(1)21 February 2020 ($T_{lock,1}$), creation of two quarantined red areas under strict lock-down in Lombardy and Veneto;(2)24 February 2020 ($T_{lock,2}$), school closure in most regions in the Northern of Italy (Lombardy, Veneto, Emilia-Romagna, Friuli Venezia Giulia, Liguria, Piedmont and part of Marche);(3)5 March 2020 ($T_{lock,3}$), school closure in the entire country;(4)8 March 2020 ($T_{lock,4}$), total lock-down area in the Northern of Italy; and,(5)10 March 2020, total lock-down area extended to all Italian regions.

All of these containment measures are implemented in the model through parameter vector $s_{0}\;=\;\left[s_{01},s_{02},s_{03},s_{04}\right]$. Since the lock-down was imposed at a distance of two days in the Northern and remaining part of Italy, we include both of them in a unique parameter ($s_{04}$), when also considering the delay in social acceptance of the restrictive intervention. Parameter $s_{1}$ represents the reduction of transmission of mild infected people due to a total ban on leaving their houses, while parameter $s_{2}$ implements the employment of PPE in hospitals. In more detail, transmission rate parameters are defined, as follows:$b_{e,lock}=b_{e}$ from day 0 until $T_{lock,1}$, $b_{e,lock}=b_{e}\cdot s_{01}$ from $T_{lock,1}$ to $T_{lock,2}$, $b_{e,lock}=b_{e}\cdot s_{02}$ from $T_{lock,2}$ to $T_{lock,3}$, $b_{e,lock}=b_{e}\cdot s_{03}$ from $T_{lock,3}$ to $T_{lock,4}$ and $b_{e,lock}=b_{e}\cdot s_{04}$ from $T_{lock,4}$ onward;$b_{0,lock}=b_{0}$ from day 0 until $T_{lock,1}$, $b_{0,lock}=b_{0}\cdot s_{01}$ from $T_{lock,1}$ to $T_{lock,2}$, $b_{0,lock}=b_{0}\cdot s_{02}$ from $T_{lock,2}$ to $T_{lock,3}$, $b_{0,lock}=b_{0}\cdot s_{03}$ from $T_{lock,3}$ to $T_{lock,4}$ and $b_{0,lock}=b_{0}\cdot s_{04}$ from $T_{lock,4}$ onward;$b_{1,lock}=b_{1}$ from day 0 until $T_{lock,1}$ then $b_{1,lock}=b_{1}\cdot s_{1}$ from $T_{lock,1}$ onward;$b_{2,lock}=b_{2}$ from day 0 until $T_{lock,1}$ then $b_{2,lock}=b_{2}\cdot s_{2}$ from $T_{lock,1}$ onward; and,$b_{3,lock}=b_{3}$ from day 0 until $T_{lock,1}$ then $b_{3,lock}=b_{3}\cdot s_{2}$ from $T_{lock,1}$ onward.

The parameter vector to estimate for Italy consists of fifteen model parameters and six interventions parameters, i.e., $p\in R^{21}$.

Tuning parameters of CRC are set, as follows:the number of samples in the parameter space is set to $N_{S}=10^{5}$ for each iteration;to perturb the parameter space, we use Linear and Logarithmic LHS according to the prior distributions shown in Table 1;the number of iterations is equal to 8; and,the number of realizations performed is set to 10, to ensure reliability of results.

Figure 2 depicts the behavior of the output variables in Italy, when parameters are equal to the mode of the approximate posterior distribution returned by CRC in one of the final realizations (see Table 1). Model simulations are shown together with the dataset, proving the adherence of the model with the data for all three variables. On the other hand, it is clear from Figure 2 that there is a significant shift between mild infections that were predicted by the model and the observed data, confirming our initial hypothesis. While using the mode vector, the estimate of the initial reproduction number $R_{0}$ is equal to 4.65, in accordance with [20]. We also estimate that containment measures have progressively reduced the presymptomatic and asymptomatic transmission by 40% in the first phase up to 90% during the total lock-down. On the other hand, the quarantine and isolation of mild infected and wearing PPE have taken to a reduction in transmission of about 85% and 80%, respectively. The presymptomatic transmission rate is estimated much higher with respect to all of the other rates, meaning that people in the presymptomatic phase play a crucial role in the COVID-19 spread.

### 3.2. CRC Results: Umbria Case

Regarding Umbria, there are only four intervention parameters to estimate (school closure and total lock-down), i.e., $p\in R^{19}$. Thus, transmission rate parameters are defined, as follows:$b_{e,lock}=b_{e}$ from day 0 until $T_{lock,3}$, $b_{e,lock}=b_{e}\cdot s_{01}$ from $T_{lock,3}$ to $T_{lock,4}$ and then $b_{e,lock}=b_{e}\cdot s_{02}$ from $T_{lock,4}$ onward;$b_{0,lock}=b_{0}$ from day 0 until $T_{lock,3}$, $b_{0,lock}=b_{0}\cdot s_{01}$ from $T_{lock,3}$ to $T_{lock,4}$, $b_{0,lock}=b_{0}\cdot s_{02}$ from $T_{lock,4}$ onward;$b_{1,lock}=b_{1}$ from day 0 until $T_{lock,3}$ then $b_{1,lock}=b_{1}\cdot s_{1}$ from $T_{lock,3}$ onward;$b_{2,lock}=b_{2}$ from day 0 until $T_{lock,3}$ then $b_{2,lock}=b_{2}\cdot s_{2}$ from $T_{lock,3}$ onward; and,$b_{3,lock}=b_{3}$ from day 0 until $T_{lock,3}$ then $b_{3,lock}=b_{3}\cdot s_{2}$ from $T_{lock,3}$ onward.

The tuning parameters of CRC are set in the same way as Italy. Figure 3 shows the result of the application of CRC to the Umbria region. The method is successful in replicating the behavior of H, ICU, and D patients, while we notice the same shift as before for mild infected people. This mismatch finds a complete confirmation in Figure 4, where the model prediction of new positive cases is in agreement with the data collected by the Italian National Institute of Health. Indeed, the peak of new cases predicted by the model and the peak of symptom onset registered by the Italian National Institute of Health are both around mid-March. The estimate of $R_{0}$ for Umbria region is equal to 5. CRC estimates similar values for process parameters for both Umbria and Italy, such as the presymptomatic rate and period and the incubation period (see Table 1). On the other hand, the probability of death (ProbDeath) is considerably lower in Umbria, since it is one of the Italian regions that registered the lowest mortality rate of COVID-19 [23].

Regarding the computational cost, CRC takes around one hour to complete ten realizations of one iteration. All the simulations are performed using Matlab (R2019a) on a Intel Core i7-4700HQ CPU, 2.40GHz 8, 16-GB memory, Ubuntu 18.04 LTS (64 bit).

### 3.3. CRA Results: Italy Case

After model calibration, we run the CRA algorithm on the Italy case, through the CRA Toolbox software [16], in order to identify those model parameters that influence most the healthcare capacity, i.e., H and ICU variables. To this purpose, we choose, as an evaluation function, the area under each one of the two curves. The lower and upper boundaries of the sampling intervals for the parameter space are fixed equal to the lower and upper 90th percentile of the final pdf estimated by CRC (see Appendix A). We perturb the parameter space with Linear LHS generating $10^{4}$ samples and we set equal to 1000 the dimension of the upper and lower tail of the evaluation function pdf, in order to guarantee a stable estimation of the conditional parameter pdfs [15]. We perform 10 realizations of the entire procedure to ensure the invariance and stability of results.

Through the CRA, we analyze two different temporal scenarios: a short one from the end of January until mid-May (stop time equal to 110) and a longer one until the end of October (stop time equal to 300). In the second case, we add more intervention parameters for asymptomatic and presymptomatic transmission, in order to take the different intervention measures established by the Italian government from the beginning of May into account. More in detail, from 4 May some businesses were allowed to reopen (parameter $s_{05}$) while from 18 May lock-down measures were almost completely lifted, maintaining only social distancing, mandatory PPE in closed places and ban of most public events (parameter $s_{06}$). Moreover, from 15 June, discos and clubs were able to open again before being shut down on 17 August, due to an increase in the number of infections, especially among younger people (parameter $s_{07}$ and $s_{08}$). Finally, we also consider school reopening from 14 September (parameter $s_{09}$), even though its consequences would be completely understandable in a few weeks.

Because these intervention parameters are not estimated with CRC, their range of variation is set to [0.1–0.9]. Through a wide perturbation of these parameters, we seek to understand the impact of non-pharmaceutical measures on the time evolution of the epidemic, and to identify a suitable scenario of the current and future situation.

Figure 5 shows the MIRI values that were returned by the CRA in the two different scenarios while in Appendix A the corresponding lower and upper parameter pdfs are reported. In the first scenario, MIRIs are similar for both H and ICU variables, which are strongly influenced by presymptomatic transmission through parameters $b_{e}$ and PresymPeriod. MIRIs also highlight the importance of parameter $s_{02}$, which represents the school closure in the Northern Italian regions about one month after the starting of the epidemic. As expected, the initial reproduction number $R_{0}$ also has strong implications for COVID-19 control in hospitals. As regards the second scenario, in both cases the most relevant parameter is $s_{07}$, confirming the fact that the opening of bars and clubs, through the attraction of large crowds, in total disregard of social distancing, has facilitated a hike in cases during the summer. Moreover, the fraction of asymptomatic people and parameters $s_{08}$ and $s_{09}$ also have quite high MIRI values. In Figure 6, a long-term prediction of the evolution of hospitalized people is shown together with the Italian data, when choosing the newly introduced intervention parameter values from the conditional pdfs $f_{p_{w}|L}$ (see Appendix A). Figure 6 also shows the evolution of new positive cases from February until now, demonstrating the initial underestimation of infected people.

### 3.4. CRA Results: Umbria Case

We repeat the CRA analysis for the Umbria region, while using the same tuning parameters of Italy. The same as above, we study two different temporal scenarios, one until May and one until October, in order to investigate the efficacy of the model in making future predictions. Figure 7 shows MIRI values returned by the CRA in the two different scenarios, while, in Appendix A, we show the corresponding parameter pdfs. In the short scenario, the results are similar to the case study of Italy, with one relevant difference. Indeed, parameter $s_{01}$, representing the initial social distancing measures, together with the school closure, is the parameter with the highest MIRI for both H and ICU. In the long-term scenario, the CRA output for Umbria is alike the Italian one, reflecting the importance of the national government decisions. Figure 8 shows model predictions for hospitalized patients and new positive cases, when the intervention parameter values not used for calibration are taken from $f_{p_{w}|L}$.

The computational cost of the CRA Toolbox is considerably lower than that of CRC, since the CRA runs ten realizations in around 10 minutes with the same computational power that is used for CRC.

## 4. Discussion

The framework proposed here, which combines Bayesian model calibration and conditional robustness analysis, provides useful insights into the spread of the COVID-19 outbreak and allows for us to evaluate multiple future scenarios on the evolution of the pandemic. Our study highlights the remarkable importance of implementing rigorous non-pharmaceutical interventions together with a responsible behavior in order to fight the threat that is represented by presymptomatic and asymptomatic people in the spread of COVID-19.

In both of the presented case studies, CRC successfully calibrates the model against the chosen observables, revealing new important findings. Our analysis shows the crucial role that is played by presymptomatic individuals in transmitting the infection. Indeed, the presymptomatic transmission rate $b_{e}$ is estimated much higher with respect to all the other transmission parameters ($b_{0}$, $b_{1}$, $b_{2}$, and $b_{3}$), meaning that infected people at this stage of disease notably speed up the contagion. The other parameter related to presymptomatic transmission is the duration of the presymptomatic period. Given an incubation period of about five days estimated by CRC, in accordance with [24], the presymptomatic phase lasts about three days, during which infected people may transmit the virus without showing symptoms. Moreover, the CRA reveals that these two parameters must be controlled in order to contain the number of people that require hospitalization and ICU. Even if PresymPeriod cannot be reduced, parameter $b_{e}$ can be mitigated through protective masks, frequent hand washing, and social distancing. According to the conditional pdf of $b_{e}$ (see Appendix A), the number of hospitalized and ICU patients can be limited by trying to keep $b_{e}$ under 1.04. This result emphasizes the relevance of a policy of population-wide testing and contact tracing in order to detect and isolate presymptomatic cases as soon as possible. Containment measures, such as swab testing and centralized quarantines, are also essential for confining asymptomatic people, estimated at about 31% of the total infections in Italy (29% in Umbria) by our calibration procedure, consistently with [24,25]. The role of asymptomatic individuals is also highlighted in the CRA analysis of the long-term scenarios. Indeed, now that the total lock-down has been lifted, detecting and isolating people without symptoms is essential in order to contain the second wave of contagion. At the beginning of the pandemic, our national health system was not prepared to face the emergency and tested only patients with flu symptoms. On the other hand, now in most regions, the number of swab tests has strongly increased and all possible suspected people are tested [26].

The time behavior of mild infections predicted by our model during the lock-down period is in agreement with [24], where it is stated that the median time between onset of symptoms and positive diagnosis ranges between two and six days. Moreover, CRC estimates an higher number of people with mild symptoms, confirming the fact that in the official data the number of mild infections is underestimated [20]. This is mainly due to the fact that the testing capacity of each Italian region was not always able to increase at a similar rate as the epidemic spread and was sometimes impaired by the lack of kits and reagents [4]. In addition, the model correctly predicts the peak of the curve of new positive cases in accordance with the Italian National Institute of Health data for Umbria region. If the model is correctly specified, then the difference between observed and model based infections of SARS-CoV-2 represents the estimate of undiagnosed and unreported cases. Furthermore, the prediction of new positive cases in comparison with the data confirms this hypothesis and shows that, through more intensive testing, we can promptly isolate positive individuals and reduce the chain of transmission. According to this prediction, school reopening may cause the rise of new infections and we are just starting to observe its impact.

Subsequently, the CRA reveals the extreme importance of the lock-down timeliness in order to reduce the epidemic peak. The Italian example, through parameter $s_{02}$, shows that acting about one month after the start of the epidemic, trying to reduce the transmission of at least 50%, is determinant for attenuating the overload of the health system’s capacity. On the other hand, a too early lock-down would have only postponed the outbreak, but not its strength [27]. While these restrictions were adopted at the beginning of March, we started seeing their effectiveness around April due to the clinical and epidemiological characteristics of COVID-19 [11]. Indeed, both the natural progression of the disease and the residual transmission during the lock-down phase, such as household transmission, have generated long delays between the beginning of restriction measures and the observation of their efficacy. This tardiness may also impair the analysis of the impact of multiple containment measures, when they are implemented within a few days between each other. Thus, in this case, also the institution of two red areas in Lombardy and Veneto, corresponding to parameter $s_{01}$ in the model, is incisive for the control of the number of hospitalized people. However, its effect is hidden by the successive restrictive measure (parameter $s_{02}$), since they occur at a distance of three days one from the other. In the Umbria example, the highest MIRI in the short scenario is the one that is related to $s_{01}$, corresponding to the initial strict movement restrictions that are imposed by the Italian government. This difference with the Italian scenario is due to the fact that the epidemic in Umbria started some weeks later compared to other Italian regions. Thus the initial measures of social distancing and school closures have been more determinant in reducing the number of infections.

Given the CRA results, we also make some model predictions on the evolution of the pandemic, while taking the different events that occurred in Italy after the lock-down release into account. Through the tuning of the more recent intervention parameters, the model is able to correctly reproduce the actual time behavior of H and ICU variables in Italy and Umbria. In agreement with the parameter pdfs that are returned by the CRA, we are currently in a phase of slow resurgence of infections, which are still bearable by the national health system.

Finally, this study has some limitations that, however, do not compromise our main conclusions. The model does not take into account the seasonality in transmission rates, as occurs with other respiratory diseases such as the influenza. Furthermore, since the Italian health system is highly decentralized, a natural improvement of this work would be to implement regional calibrations as we did for Umbria, taking into account the initial banning of the cross-regional movement [28]. The introduction of age classes would also help in refining model predictions, since the progress of the disease is strongly dependent on the age of the patient.

## 5. Conclusions

In this article, we apply a new Bayesian calibration method called CRC to model the COVID-19 scenario in Umbria and Italy during the lock-down phase. The model reveals the high transmissibility of the infection from asymptomatic and presymptomatic individuals and the real evolution of mild infections in the early stages of the pandemic, clearly underestimated in the official data. Then, the CRA analysis highlights the importance of the initial lock-down measures and of a policy of massive testing in order to contain the number of hospitalizations and allows us to make accurate predictions on the evolution of the disease in Umbria and Italy.

## Figures and Tables

**Figure 1 biology-09-00394-f001:**
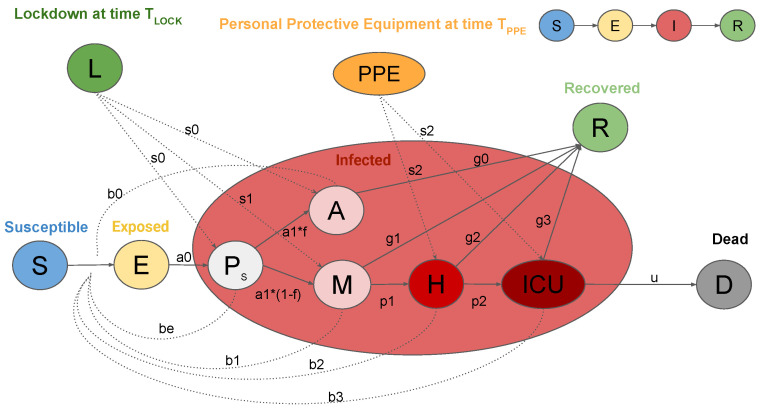
Graphic representation of the SEIRL model. Clinical stages of infection are: Susceptible (*S*), Exposed (*E*), Presymptomatic ($P_{S}$ ), Asymptomatic (*A*), Recovered (*R*), Mild infection (*M*), Severe infection (*H*), Critical infection (ICU), and Dead (*D*). Control measures: Lock-down measures (*L*) and personal protective equipment (PPE).

**Figure 2 biology-09-00394-f002:**
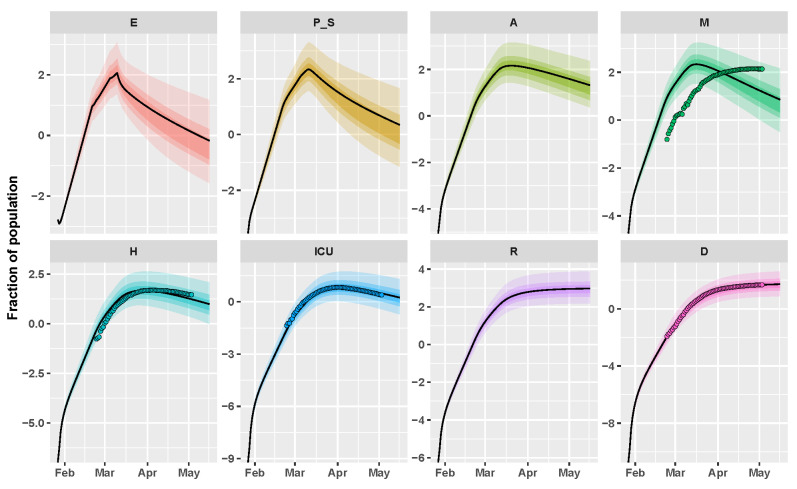
Italy example. Time behavior of state variables when the parameter vector is equal to the final mode vector computed by CRC (black line); dots represent the data available at [21]. Data and simulations are in log-scale, normalized over the whole Italian population (∼60 million) and multiplied by $10^{5}$. The colored area reproduces the variation of the temporal behavior when parameters vary between the 60th, 70th, and 90th percentile of their corresponding conditional pdfs (see Appendix A). Time starts from 27 January 2020.

**Figure 3 biology-09-00394-f003:**
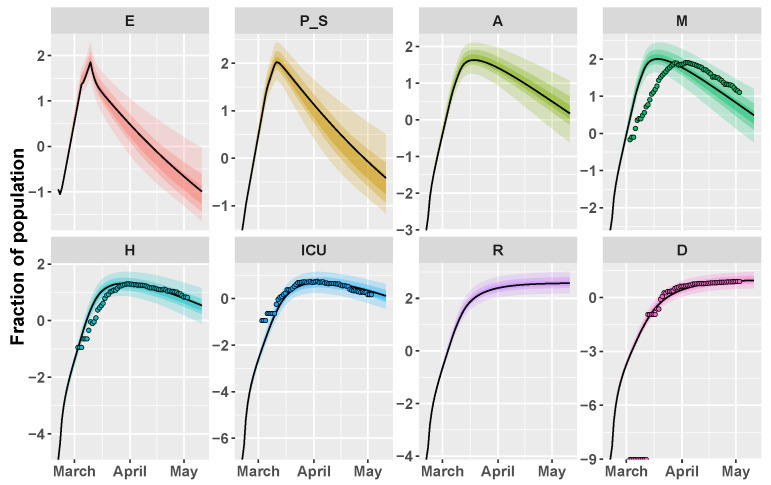
Umbria example. Time behavior of state variables when the parameter vector is equal to the final mode vector that is computed by CRC (black line); dots represent the data available at [21]. Data and simulations are in log-scale, normalized over the whole population of the region (∼882,000) and multiplied by $10^{5}$. The colored area reproduces the variation of the temporal behavior when parameters vary between the 60th, 70th, and 90th percentile of their corresponding conditional pdfs (see Appendix A). Time starts from 21 February 2020.

**Figure 4 biology-09-00394-f004:**
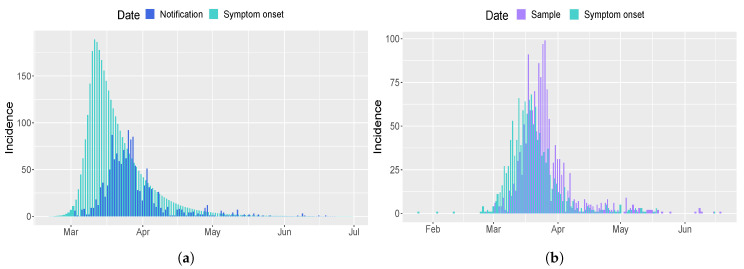
Comparison of new positives cases for Umbria region. (**a**) Epidemic curve estimated by the model (light blue) in comparison with the observed data of the Italian Civil Protection Department (blue) [21]. (**b**) Epidemic curve based on date of sample (violet) and symptom onset (light blue) estimated by the Italian National Institute of Health.

**Figure 5 biology-09-00394-f005:**
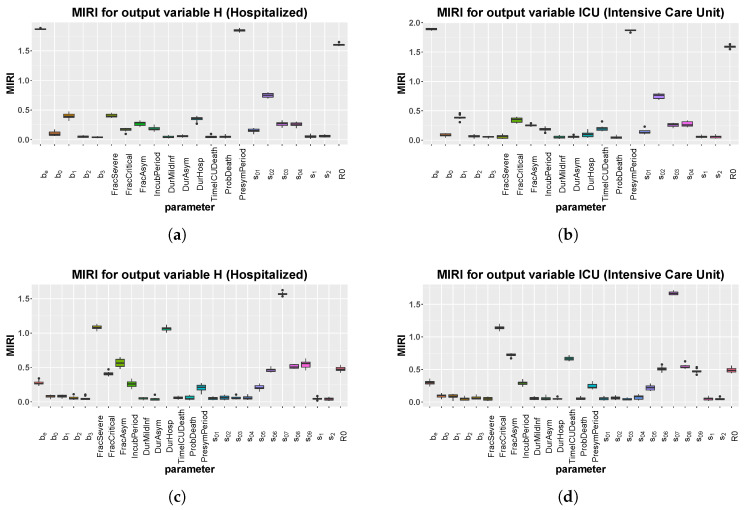
Italy: boxplot of the 10 realizations of the MIRIs in output from the CRA for all model parameters. (**a**) Evaluation function: area under the curve of H. Stop time: 110. (**b**) Evaluation function: area under the curve of Intensive Care Unit (ICU). Stop time: 110. (**c**) Evaluation function: area under the curve of H. Stop time: 300. (**d**) Evaluation function: area under the curve of ICU. Stop time: 300.

**Figure 6 biology-09-00394-f006:**
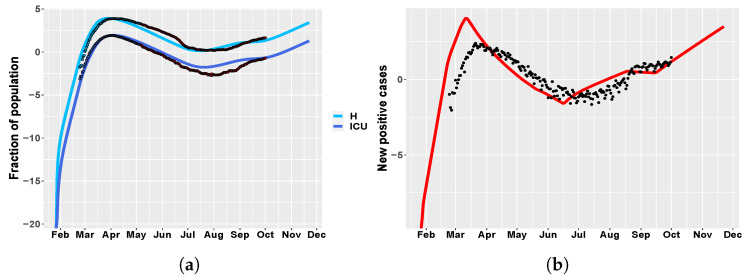
Italy. (**a**) time behavior of hospitalized (light blue) and ICU (blue) patient in comparison with data of the Italian Civil Protection Department [21]. Data in red represent the second portion of the available dataset, from 4 May until 1 October 2020, which were not employed in model calibration. The newly introduced intervention parameters are set to: $s_{05}=0.11$, $s_{06}=0.13$, $s_{07}=0.27$, $s_{08}=0.22$, $s_{09}=0.3$. (**b**) Italy: time evolution of new positive cases in comparison with the data in [21].

**Figure 7 biology-09-00394-f007:**
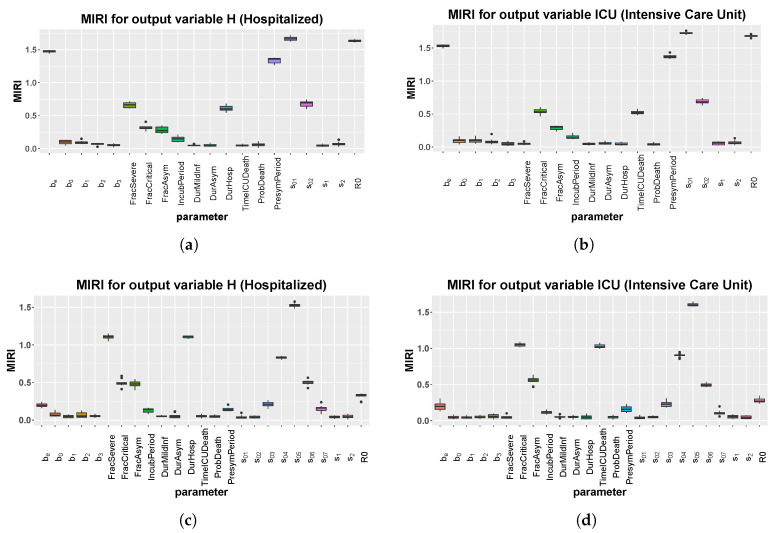
Umbria: boxplot of the 10 realizations of the MIRIs in output from the CRA for all model parameters. (**a**) Evaluation function: area under the curve of H. Stop time: 110. (**b**) Evaluation function: area under the curve of ICU. Stop time: 110. (**c**) Evaluation function: area under the curve of H. Stop time: 250. (**d**) Evaluation function: area under the curve of ICU. Stop time: 250.

**Figure 8 biology-09-00394-f008:**
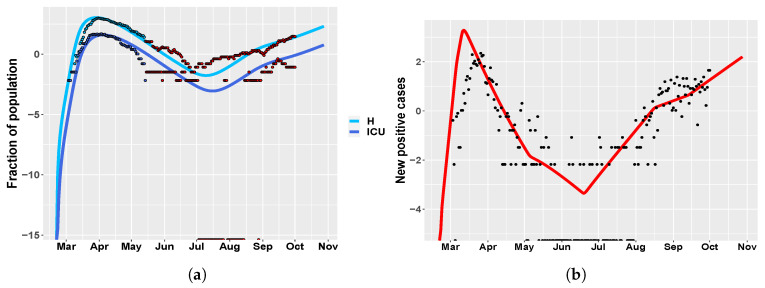
Umbria. (**a**) time behavior of hospitalized (light blue) and ICU (blue) patient in comparison with data of the Italian Civil Protection Department [21]. Data in red represent the second portion of the available dataset, from 4 May until 1 October 2020, which were not employed in model calibration. The newly introduced intervention parameters are set to: $s_{03}=0.15$, $s_{04}=0.15$, $s_{05}=0.27$, $s_{06}=0.22$, $s_{07}=0.25$. (**b**) Umbria: time evolution of new positive cases in comparison with the data in [21].

**Table 1 biology-09-00394-t001:** Conditional Robust Calibration (CRC) results for Italy and Umbria. The second column shows the initial prior distribution of parameters. The initial range of transmission rates is taken from [10]. The fourth and fifth columns show, respectively, the mode vector of $f_{P|P_{O,\epsilon }}$ in one of the 10 final realizations for Italy and Umbria. In order to avoid integration errors, we suppose that the presymptomatic period (PresymPeriod) is a percentage of the incubation period (IncubPeriod).

Parameter	Prior	Italy $p_{\mathrm{mode}}$	Umbria $p_{\mathrm{mode}}$
$b_{e}$	log-U(0.001,3)	0.98	1.2633
$b_{0}$	log-U(0.001,3)	0.0292	0.0115
$b_{1}$	log-U(0.001,3)	0.0585	0.0052
$b_{2}$	log-U(0.001,3)	0.0249	0.0045
$b_{3}$	log-U(0.001,3)	0.0066	0.0169
FracSevere	log-U(0.01,0.4)	0.1066	0.1131
FracCritical	log-U(0.01,0.3)	0.0944	0.1308
FracAsym	U(0.1,0.6)	0.3127	0.2916
IncubPeriod	U(4,14)	5.7009	5.4046
DurMildInf	U(2,80)	9.2534	9.9375
DurAsym	U(2,30)	19.8963	10.4581
DurHosp	U(2,90)	16.2593	12.0868
TimeICUDeath	U(2,70)	4.8568	5.7772
ProbDeath	U(1,90)	88.6568	27.0937
PresymPeriod	log-U(0.5,0.9)	0.7243	0.7248
$s_{01}$	log-U(0.5,0.9)	0.5922	0.5732
$s_{02}$	log-U(0.4,0.9)	0.5035	0.0955
$s_{03}$	log-U(0.3,0.7)	0.3982	-
$s_{04}$	log-U(0.05,0.5)	0.1031	-
$s_{1}$	log-U(0.1,0.9)	0.1452	0.2360
$s_{2}$	log-U(0.1,0.9)	0.1842	0.2481

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
