# Peer review of "Mathematical Modeling and Robustness Analysis to Unravel COVID-19 Transmission Dynamics: The Italy Case†"

_biology, 2020, doi:10.3390/biology9110394_

Round 1

Reviewer 1 Report

This is an elegant model that processes data in the public domain to assess effectiveness of the distancing measures. I am happy to provide approval for this timely work when following comments are addressed.

  1. I strongly suggest extensive revision of language. All acronyms must be defined, all sentences simplified, use of simple tenses preferred over subjunctives, and scientific writing methods pursued in the revision.
  2. All equations and their variables should be explained.
  3. Since this manuscript is targeted at those who read in English, text as well as figures should be revised to the effect.
  4. In Figs 1, 7, and 8, grayscale and fontsize made certain parts of the figures illegible, which should be addressed.
  5. Since the data are from the public domain, the authors should consider sharing of the prepared data by means of their github repo.

Reviewer 2 Report

Eq. (1): There is no need to introduce class D. A death rate in eq. for ICU is sufficient.
              It is strange that there is no 'birth rate' for the system (1): the compartment for S
              can only decrease ?!

Eq.(2) : Details are missing. Is R_0 computed by eigenvalues of linearization matrix? This is important, since there exist different definitions of R_0.

The Eqs. (3) make no sense at all. One can write this assignment in a programming language, but not as a mathematical equation!!

System (1) is not analysed: what is the endemic equilibrium? Is it attractive ? ETC...

Reviewer 3 Report

see attached file

Round 2

Reviewer 1 Report

All my concerns have been addressed. Whereas any project will benefit from further improvement, I am happy to approve this manuscript for the next step in publication. Congratulations to the authors who are both well qualified and well placed for such a study.

Reviewer 2 Report

I am satisfied with the changes made